# IoT-based Retrofitting Model for Data-driven Water Flow Monitoring

Nilesh Bawankar, Ankit Kriti, Himani Pande
International Institute of Information Technology, Hyderabad

30th October, 2023

**Abstract**

The monitoring of water flow plays a crucial role in detecting leaks and minimizing water wastage, ultimately enhancing the management and conservation of our valuable water resources. Addressing this challenge calls for the development of an efficient and sustainable water management system. We introduce a solution based on the Internet of Things (IoT) that focuses on retrofitting existing analog water meters with readily available off-the-shelf electronic components to make them smart. Real-time data collection and analysis are seamlessly executed through edge computation using Raspberry-Pi. The water meter images are captured by Raspberry-Pi camera and are processed to estimate the meter readings using a machine learning (ML) model. To evaluate the developed solution's reliability and performance, extensive field deployments spanning several months are conducted, enabling the characterization of water usage patterns and fault detection across diverse locations. Additionally, the integration of energy harvesting capabilities into the system reduces maintenance costs and promotes eco-friendly energy practices. In summary, this solution presents a comprehensive and effective approach to achieving efficient and sustainable water management.

## 1   Introduction

The conventional method of manually reading analog water meter readings to calculate the water consumption trend is cumbersome and expensive. This approach is also incapable of effectively managing sustainable water supplies, as it needs accurate monitoring techniques that enable the consumer to know the level of water usage in real-time. The traditional analog water meters have a long life, and removing them for digitization is a waste of resources. Although digital water meters have been introduced in recent times and are used in workplaces like government institutions, and hospitals, they are very expensive. Also, the digital water meters themselves do not give any inference or do not do any analytical analysis of the water consumption patterns. A smart device for water monitoring can make users reduce their use of water to conserve it.

Retrofitting can enable more efficient water resource management by providing real-time data on water consumption, without the need for replacement as in the case of digital water meters, helping utilities and municipalities detect leaks and reduce water wastage. The integration of IoT into this will prove to be beneficial for the Smart Cities where automation is the prime concern.

In this project, an IoT-based sensor node is designed and deployed in the field to take water meter images, convert them into digits, and send them to the cloud. A retrofitting approach is used which does not tamper with the existing meter in any physical form. It is assumed that the meter does not have any pulse output and relies completely on the conversion of the images to digits. The designed node can provide real-time data with high temporal resolution and is equipped with lighting to enable readings even at night. This way the meter readings can also give accurate estimates of derived parameters such as flow rate. A simple ML algorithm is used to recognize the digits from the meter image, which requires low processing and is implemented at the node itself. The performance of the ML algorithm is further improved by using specific constraints related to the water meters. The proposed approach is evaluated based on the data of over 10,000 images collected from the field deployment of 10 days. The sensed readings are stored on a cloud-based platform for analysis purposes. The proposed model was deployed at the IIIT-H campus for real-time analysis and data collection. The image processing algorithm achieved an accuracy of 97.69% for digit recognition.

## 2  Goals

1.  Enable analog water meters to collect and transmit real-time data on water consumption, providing up-to-date information to utility companies and consumers.

2.  Develop a cost-effective retrofitting solution to make the conversion of analog meters to smart meters.

3.  Utilize machine learning and data analytics to provide insights into water consumption patterns, allowing for better resource management and conservation strategies.

4.  Enable remote monitoring and control of water meters through IoT, reducing the need for physical meter reading and enhancing human operational efficiency [1].

## 3  System Architecture and Design

### 3.1  Hardware

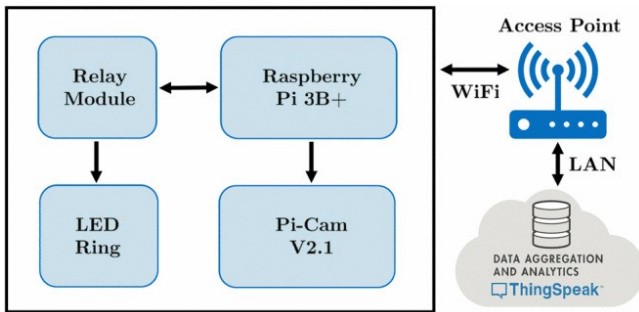

Figure 1: Block architecture of the model

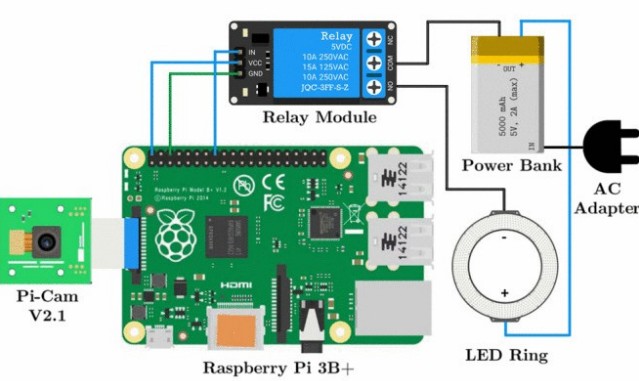

Figure 2: Circuit diagram of the model

Figs. 1 and 2 show the block architecture and circuit design of the proposed model. The model comprises a Raspberry-Pi 3B+ microprocessor [2], a Raspberry-Pi V2.1 camera module, an LED ring for illumination, and an active-high relay module to control the LED. The hardware is powered by a Li-ion power bank which is connected to AC mains for charging. The power bank enables the device to function without interruption, even if the primary AC power supply is unavailable temporarily. The model is also equipped with a lighting feature, such that the readings can be obtained even at night when there is no ambient light available. The lights are controlled using an active high relay module to operate only when the camera captures an image. This feature helps in extending the battery life of the model. The numeric values are extracted from the images of the water meter dial captured by the camera. The microprocessor executes the ML-based image

processing algorithm to detect the reading on the meter. This reading is transmitted in real-time to ThingSpeak [3] using a WiFi hotspot. ThingSpeak is a cloud-based IoT platform for aggregating and processing data. The POST method of the HTTP protocol is used to write data on the ThingSpeak server. This setup can sense the reading at a high frequency, even when the flow is at its peak. Such high-frequency information can be used to derive valuable insights about the community's consumption patterns and timely detection of leaks or faults.

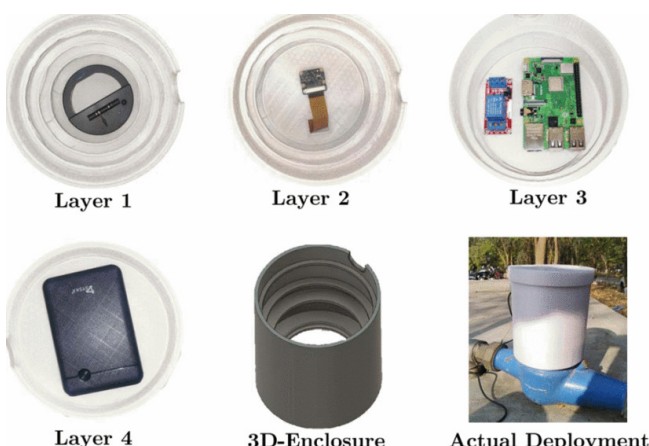

Figure 3: 3D structure and deployed model

Fig. 3 shows the structure of the developed enclosure. It is a 3D-printed multi-layer structure made up of Polylactic Acid (PLA) material which offers protection against various weather conditions in outdoor deployment. It consists of a four-layered stack that separates the various hardware components for comfortable placement. The first layer (bottom-most) consists of an LED ring for providing adequate illumination to capture good-quality images. In the second layer, the camera module is placed facing the dial of the meter. The camera is configured at a focus of 4 cm with a maximum resolution of 3280 × 2464 and pixel size of 1.12 × 1.12 μm. The Raspberry-Pi microprocessor is placed at the third layer. It is interfaced with an active high relay module using the Raspberry-Pi GPIO pins to control LED switching. At the fourth layer (topmost), the power bank is placed. The whole setup is mounted on top of the analog water meter without altering or tempering the analog meter in any sense. It is important to note that the discussed water meters are ISI-marked [4], a standards-compliance mark for India's industrial products provided by the Bureau of Indian Standards (BIS). Hence, any technological intervention made by physically altering the meters would lead to a loss of standardization.

| Component | Use |
|---|---|
| Raspberry-Pi 3B+ | Used to perform the computation related to digit recognition |
| Raspberry-Pi Camera v2.1 | Capture the image of the analog water meter reading |
| Relay Module | Controls the operation of LED ring light |
| LED Ring Light | Provides illumination for image capture process |
| Power Bank | Provides power to the retrofit system in case of temporary AC supply outage |
| AC-DC Power Adapter | Primary power supply for the retrofit system |

## 3.2   Software

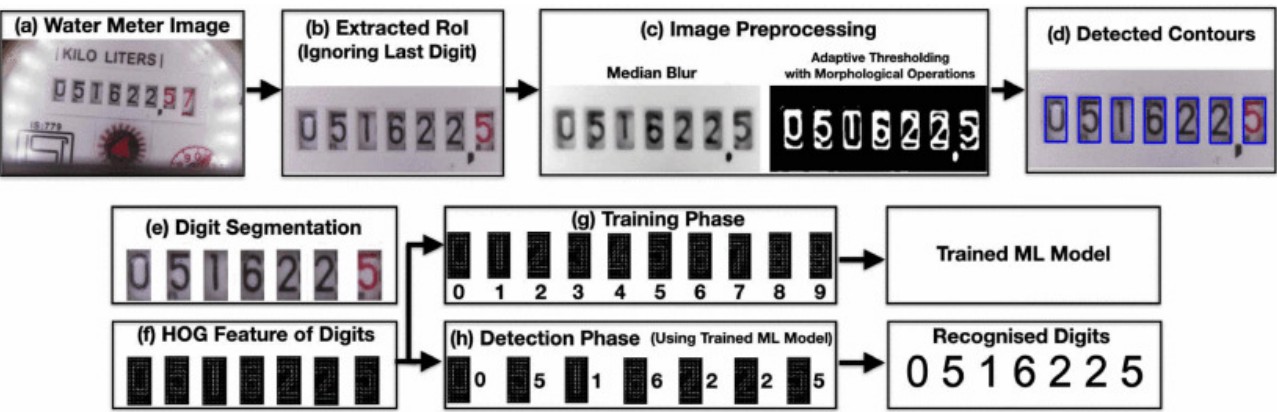

Figure 4: Algorithmic pipeline for the proposed method. Notice that the last digit of the meter is discarded due to digit image ambiguity. Both the training and detection phases have been shown. In the training phase, we train the ML model on digit features and their corresponding labels. In the detection phase, each segmented digit image feature is predicted using the trained ML model. (Best viewed on screen)

With the help of the above-described hardware set up on the water meter, two separate datasets were created, one for training the model and the other for analyzing the volume flow and its flow rate. As explained in the hardware section, the camera position is fixed, and the orientation of the images remains the same for all captured images. We specified the coordinates where the meter reading was present and extracted the region of interest (RoI). The resultant RoI image is shown in Fig 4(b). As shown in Fig 4(a), the meter reading is tilted towards the right side. In order to straighten the RoI with respect to the specified width and length, the perspective transformation was used. The transformation matrix is calculated from the manually selected source and destination coordinates. Using these transform coefficients, warp perspective [5] was applied to get the transformed image. This step helps us to form nice rectangle-shaped contours.

As the next step, all the digits in the complete reading are segmented into individual digit images and are stored separately as per their numerical value. This makes the dataset suitable for supervised learning. These digit images were further used to train the ML model. The dataset consisted of digits from 0 to 9. These digits were collected from the meter images captured using the Raspberry-Pi camera for 10 days, out of which 7 days of data was used to analyze the water flow and the rest for training. After creating the digit dataset, water meter images were collected to study the volume flow and its corresponding rate with which the water is flowing. For this purpose, Raspberry-Pi captured meter images for around two days at every minute, which resulted in a total of 10, 508 images. These images were then analyzed with the algorithm's help to get the volume flow and its corresponding flow rate.

### 3.2.1   Training

The digit dataset, explained in the dataset section, was used to train an ML model to classify the given digit image from 0 to 9. As this is a supervised learning problem in ML, a classification-based method was used to train the model. There are several classification algorithms in this particular area. However, we chose a tree-based method to solve the problem at hand. In this paper, Random Forest (RF) [6] classifier was used to train on the digit dataset. RF is a supervised learning algorithm that combines multiple decision trees and is trained together with bagging [7]. The bagging method uses the idea that the combination of several weak learning models increases the overall result. To explain the RF model more straightforwardly, it can be said that it combines multiple decision trees and merges the results obtained by them to get a more accurate and stable prediction. The idea behind choosing this particular classifier is to reduce the number of hyperparameters while training the model. It is also one of the most used classifiers for these problems because of its simplicity and diversity. The training procedure follows these steps:

1. For each digit in the dataset, the digit's Histogram of Oriented Gradients (HOG)-based features [8] were computed, which is a two-dimensional matrix as shown in Fig 4(f). This matrix was flattened and converted to a one-dimensional feature vector with size n × 1. As the image height and width remain the same for all the digit images, we always get a one-dimensional n × 1 feature vector for each digit image. At the end of this step, we have m × n sized data matrix M where m is the number of samples in the digit dataset.

2. In the second step, the corresponding labels were collected for each digit image in the dataset. The label ranges from 0 to 9. After this step, we have m × 1 label vector.

3. The data matrix M was split into two parts: training and validation with the ratio 80:20, respectively. This is a standard paradigm used in ML methods to test the generalization of the trained model. The RF model is trained on training data only and tested on validation data to find the error rate. The training data was used to train the RF model as a classification-supervised learning problem. The RF model's input is the training data matrix M (training part of M) and the corresponding label vector. The training phase is shown in Fig 4(g). Finally, this trained model was saved for further detection tasks.

### 3.2.2   Detection

To detect the digits of the water meter we present a 4-step structure: i) RoI Extraction, ii) Image Preprocessing, iii) Digit Image Segmentation, iv) Digit Recognition and Correction.

1. **Region of Interest (RoI) Extraction:** The specific region in the image, which consists of the digits, i.e., the object area, is manually extracted by inputting the RoI's coordinates in the algorithm. We have leveraged the fact that the camera is permanently fixed in one position, and hence setting the coordinates once is sufficient to get the fixed RoI. Notice that the last two digits of the meter reading are decimal places. We have not included the last digit in RoI as this part is the most ambiguous and sometimes even hard for humans to detect the reading. Again, as mentioned in the dataset section, we use perspective transformation to straighten the RoI.

2. **Image Pre-processing:** Recognizing the location of digits from the RoI is a challenging task. The RoI can be noisy and blurred because of the dusty environment and the presence of dew on the water meter. Hence we need to pre-process the image beforehand to find the location of the digits. We use the following methods to pre-process the image: i) grayscale [5], ii) median blur [5], iii) adaptive thresholding [5]. Initially, the RGB-colored image is converted to a grayscale image to reduce the complexity and computation overhead: from a 3-dimensional pixel value (R, G, B) to a 1-dimensional value. Later on, the grayscaled image is median blurred to smoothen out the edges. Hence, all the high-frequency components (noise) of the image will be removed. Finally, adaptive thresholding followed by morphological operation (dilation) was performed on the blurred images to separate desirable foreground image objects (digits) from the background based on the difference in each region's pixel intensities. After this step, from Fig 4(c), it can be noticed that a closed curve is successfully formed around the digits, which will further help in determining the location of the digits.

3. **Digit Image Segmentation:** The preprocessed image was used to find the location of the digits in the RoI. We have used the closed curves formed outside the digits on the water meter to determine the location. In any image, contours are curves or continuous lines that join all the continuous points, having the same color or intensity, to bind an object's complete boundary in the image. As in our case, the curves are formed around digits, finding them will provide the digit's locations. The contour retrieval mode was set to retrieve only the outer contours, so only the outermost is given, in case we have one contour enclosing another (like concentric circles). The contour approximation method is set to remove all redundant points and compress the contour to save memory. However, there are other contours as well in the preprocessed image that are not formed around the digits and they are discarded based on the contour area. The contours formed on the RoI image are shown in Fig 4(d). After image-processing and detecting the selected region, each digit present in that contour is segregated and extracted. This was made possible cause the contour stores the coordinates information as well.

This process is called image segmentation, in which the image is partitioned into different regions based on a common feature, in our case, the digits in the selected region. As each contour is formed around digits only, the contour coordinates are sorted from left to right based on their position. Fig 4(e) shows the segmented digit images. In the next step, each extracted digit image is detected using the trained ML model.

4. **Digit Recognition and Correction:** The last step of the proposed system is the recognition process for the meter reading digit images obtained in the previous step. The digit images that were segmented with the contour method are now passed to the trained RF model that we created while training. The digit image features were computed using the HOG feature extractor whose output is n × 1. As shown in Fig 4(h), the RF model's input is one digit image feature at a time, and the output is the corresponding digit. As the digits may be detected wrongly, two-digit correction mechanisms as post-processing are applied to the number collected after combining the predicted digits. These mechanisms are defined as follows:

   (a) As this is continuous real-time chronological data, it is assumed that the current value must be greater than or equal to the previous value. This helps us to mitigate the common detection error that occurred while detecting the digits.

   (b) Another assumption is made that the flow of the water can not increase suddenly by a huge number. Leveraging this fact, the digits detected are adjusted based on the previous flow.

## 4 Addressing Challenges

1. Wi-Fi network availability for deployment at remote locations
   The retrofit water meters were deployed in various parts of the IIIT-Hyderabad Campus where there was the issue of Wi-Fi network coverage. For these specific locations, the retrofit nodes are deployed with portable LTE Wi-Fi hotspots [9].

2. Use of Raspberry-Pi Camera v2.1 for high definition images
   The camera module used in the retrofit devices in its earlier stages was Raspberry-Pi v1.3, which resulted in blurred out-of-focus images. Hence, Raspberry-Pi Camera v2.1 was used because of its changeable focus and 8-megapixel image quality.

3. Hermetically sealing the retrofit device on the analog water meter
   In the rainy seasons, it was noticed that there was some accumulation of dust and dew inside the retrofit casing which hampered the image quality of the meter readings. To solve this issue, morphological operations are performed on the extracted RoI to make digit detection easier. As a precautionary measure, during the installation of the device, the analog water meter is cleaned and the retrofit device is hermetically sealed on the meter.

4. Use of computational light ML model
   The use of computational heavy CNN model [10] required a continuous power supply which resulted in the device turning off after some time even on battery power, during a temporary power outage. Hence, use of computational light ML model was used.

5. Use of post-processing on the detected reading
   The detected readings showed a sudden increase or decrease in value due to wrong detection. Hence, to solve this specific constraints related to the water meter were implemented which also improved the performance of the ML algorithm.

# 5   Performance Evaluation and Testing Results

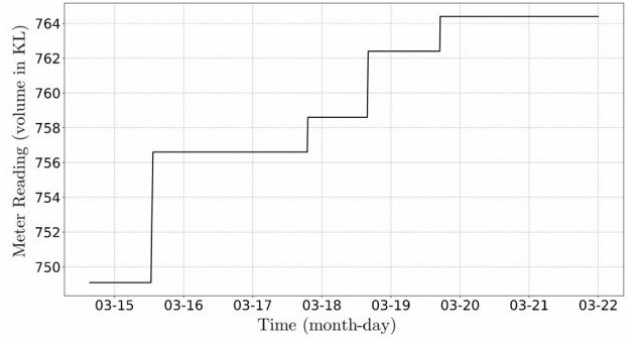

Figure 5: Plots of meter reading (volume in KL) w.r.t time, assuming 0000 hrs as the start of the day on the time axis.

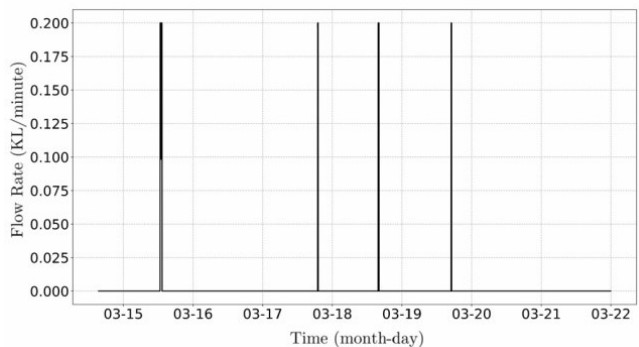

Figure 6: Plots of Flow rate (KL/minute) w.r.t time, assuming 0000 hrs as the start of the day on the time axis.

The developed model was deployed for ten days on a water meter at the pump house of IIIT-Hyderabad campus. Water from this pumping station is delivered to a residential area of forty families where it is stored in the overhead tanks. During the initial three days, the captured images were used for training the ML model. The trained model was later used for the next seven days to collect the flow data in real-time. The HOG-based image feature extraction was done using skimage [11] The orientation value was set to 9 with 8 pixels per cell and two cells per block. The RF classifier was implemented using Scikit Learn, a popular Python-based ML library. The criterion hyperparameter of the RF classifier, which measures the quality of split while training was set to entropy. For all kinds of image pre-processing, e.g., conversion to grayscale, median blurring (with window size 15), and adaptive thresholding, the popular computer vision library OpenCV [12] was used.

Figs 5 and 6 show the sensed meter readings and flow rate w.r.t time for the retrofitted water meter, respectively. The time axis is plotted such that every tick marks the beginning of the day with 0000 hrs. Therefore the time elapsed between two consecutive ticks is 24 hrs. Every pumping instance is marked by a rise in the meter reading as well as the flow rate. The meter reading remained constant, and the flow rate remained zero for the duration when the pump was not running and, consequently, no water was flowing. It is observed that the water was pumped four times in the period of observation. The duration for which the water is usually pumped is significantly less when compared to the total duration of observation. Hence, although the pumping duration is a few tens of minutes for every instance, it appears as a sudden step in the meter reading or a sharp peak in the flow rate. This is a valid observation because the campus has an automated pumping system. The water is pumped to the residential area's overhead tanks only when the tank's levels dip below a certain threshold. It is worth observing from Fig 6 (Flow rate plot) that the flow rate does not remain constant for the entire duration of water flow; instead, it usually keeps varying between 0.1 KL/minute and 0.2 KL/minute. It means that there is no deterministic linear relationship between the duration and the volume of water flow. Instead, it is governed by the entirely random demand for water.

## 6   Concluding Remarks and Avenues for Future Work

This report presented an IoT-based method for retrofitting analog water meters to convert them into smart meters. An image of the analog meter is captured at a high frequency to extract the meter reading in real-time. The RF learning method is used to perform image classification for extracting the meter readings at the edge. The sensed readings are stored on a cloud-based platform for analysis purposes. The proposed model was deployed at the IIIT-Hyderabad campus for real-time analysis and data collection. The image processing algorithm achieved an accuracy of 97.69% for digit recognition. Application-specific post-processing mechanisms achieved a low VER of 4.49% and low RMSE of 0.0361 KL. The measurements done over ten days clearly show the effectiveness of the deployed system in generating flow volume and flow rate data. In the future, it is intended to expand the deployment over all the water meters on the campus. Deep learning-based image processing techniques can also be developed to improve digit recognition further. The collected data can generate water consumption patterns in the community and help in the timely detection of leaks/faults.

## 7   Availability

1. Source Code

2. Demonstration Video

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
