# OpenReview forum: "IoT based Retrofitting Model for Data-driven Water Flow Monitoring"
_helsinki.fi/ESPC/2023/Competition — ESPC 2023 LongPresentation_

### Official Review · Reviewer_ypTN · 2023-11-15

**Rating:** 3
**Confidence:** 2

**Summary:**

Authors suggest IoT solution for water flow monitoring, in the form of external device attached to the analog water meter. This device is able to capture, extract, and send the measurements to the cloud for further analysis. Random Forest ML method was used for digits recognition. System is fully designed, implemented, and evaluated.

**Strengths:**

Authors present nice use case where IoT and image processing could aid reading the measurements from already installed analog devices and reduce labour and human error.
- System is fully designed, implemented, and tested.
- The report is nicely written.

**Weaknesses:**

I would suggest presenting the model performance results in the training phase sections.

---

### Official Review · Reviewer_7C8h · 2023-11-16

**Rating:** 4
**Confidence:** 3

**Summary:**

The authors present a novel approach to address the limitations of conventional analog meters and the high cost of digital meters. Their solution can be leveraged to monitor and gather insights on water consumption. This is a very vital step towards the broader goal of optimizing / reducing the water consumption.

**Strengths:**

- The problem and the approach to solve the problem is presented in a precise and concise manner in the report and also in the video.
- The solution is elegant, and it highlights the challenges that one might encounter in practical deployment.
- The design takes into account a wide range of practical deployment issues, such as batteries to account for the possibility of power cuts, ensuring that the standardization is not lost, and the use of LTE hotspots to address the coverage of Wi-Fi in their campus.
- The intricacies of the data processing pipeline along with the approaches to address the challenges are presented in sufficient detail. For instance, when recognizing the digits, the authors correctly assume that the meter readings are monotonically increasing and the change in the meter readings is bound by the pipe capacity.

**Weaknesses:**

There are no major weaknesses as such. The following suggestions might be useful to further strengthen this work.
- Example messages that are sent to the thinkspeak servers.
- The use of pi-zero or equivalent devices, which have a lower energy consumption than a Rapsberry Pis.
- The source code documentation can be improved.

---

### Official Review · Reviewer_J4ur · 2023-11-18

**Rating:** 2
**Confidence:** 3

**Summary:**

This project presents an IoT-based solution for water consumption monitoring. The project uses a Raspberry Pi for taking images and executing ML methods to accurately show/transmit the water consumption levels.

**Strengths:**

The project presents a solution for water meter monitoring that is useful for developing countries. The project shows the application of ML methods for classifying digits/numbers taken by RPi’s camera.

**Weaknesses:**

The project does not specify how many images are taken in a day. Indeed, continuously capturing images and applying ML would result in a considerable amount of power consumption. The project could also define a strategy (considering user behaviors) for taking and analyzing images.
In addition, the project does not talk about when using the solution at large scale. For example, how many of those smart monitoring is needed to cover the whole city of Hyderabad?
The risks of wireless connection have also not been discussed. Is Wi-Fi available everywhere in the city of Hyderabad to transmit data to the cloud?
In addition, the lifetime of the RPi has not been considered/discussed. How are those devices controlled and maintained?

---

### Official Review · Reviewer_ZLKZ · 2023-11-18

**Rating:** 3
**Confidence:** 3

**Summary:**

In this report, the authors have presented a prototype that retrofits existing analog water meters with their developed smart meters to monitor the water flow.
This prototype is built using a Raspberry Pi whose major external components are a camera and an LED Ring.
Images collected by the Pi are processed further to extract the meter readings as a digit dataset which is then fed to the ML algorithms

**Strengths:**

*Their system is deployed at various sites in their university campus
*The machine learning model has to be transformed to a computationally light version to be able to execute on the Raspberry Pi with low battery requirements
*Real-world dataset is used to train the ML model
*The solution is cost-effective and an efficient way to make existing analog meters a little smarter.

**Weaknesses:**

*Accuracy of the ML model is not discussed (it is briefly mentioned as 97.69\% in the conclusion)
*The battery consumption of Raspberry should be discussed.